# EXPLORING CONNECTIONS BETWEEN MEMORIZATION AND MEMBERSHIP INFERENCE

## ABSTRACT

Membership inference (MI) allows adversaries to query trained machine learning models to infer if a particular data sample was used in training. Prior work has shown that the efficacy of MI is not the same for every sample in the training dataset; they broadly attribute this behavior to various data properties such as distributional difference. However, systematically analyzing the reasons for such disparate behavior has received little attention. In this work, we investigate the cause for such a discrepancy, and observe that the reason is more subtle and fundamental. We first provide empirical insight that an MI adversary is very successful with those samples that are highly *likely to be memorized*, irrespective of whether the sample is from the same or a different distribution. Next, we provide a game-based formulation which lower-bounds the advantage of an adversary with the ability to determine if a sample is memorized or not, under certain assumptions made about the efficacy of the model on the memorized samples. Finally, based on our theoretical results, we present a practical instantiation of a highly effective MI attack on memorized samples.

## 1 INTRODUCTION

Advances in machine learning (ML) are enabling a wide variety of new tasks that were previously deemed *complex* for computerized systems. These tasks are powered by models which are trained on large volumes of data, that often fall under the category of being sensitive or private, as it is collected from a variety of sources. For example, data used to customize (or fine-tune) large language models can often be sensitive (Carlini et al., 2021; Zanella-Béguelin et al., 2020). Hence, understanding and explaining the privacy risks of the data used to train these models is an important problem that need to be solved before widespread adoption of these models.

Several prior works (Shokri et al., 2017; Yeom et al., 2018) have successfully established that such models are susceptible to privacy attacks, such as membership inference (MI), that aim to infer if specific data-points are used during their training. Even more concerning is that they have shown that the efficacy of MI is not the same for every sample in the training dataset. Unfortunately, the problem of explaining this discrepancy has received less attention. Only recently, researchers have proposed techniques to measure the susceptibility of attack per sample (Carlini et al., 2022a; Ye et al., 2022), and attributed the behaviour of disparate risks with coarse relation to distributional difference (Kulynych et al., 2019). Consequently, out-of-distribution (OOD) samples which are part of the training dataset were deemed to be at a higher risk compared to other samples.

In this work, we first systematically analyse the correctness of the above reasoning using representative techniques from OOD detection (Hendrycks & Gimpel, 2016; Liu et al., 2020) and MI literature (Carlini et al., 2022a). Our empirical observations reveal that the relationship between OOD samples and higher MI risk is *not* straightforward (§ 3). The reasoning is more subtle and fundamental. We bridge the gap in our understanding of this problem and provide reasons for varying MI risk among training data-points. We demonstrate that MI and OOD samples are connected via the susceptibility of *memorization* of these samples Feldman & Zhang (2020); Brown et al. (2021). That is, we show that an adversary is highly successful in predicting the membership of those samples that are likely to be memorized, irrespective of the distribution from which the data-point is sampled. Moreover, as shown in previous work by Feldman (2020), and demonstrated in our evaluation, it is

the OOD samples that have a higher tendency of being memorized by the model, thereby misleading previous work in concluding that distribution distance increases the susceptibility to MI.

Next, we formalize a new game-based definition for MI, and present connections between memorization bounds from Brown et al. (2021) and the advantage of an MI adversary. We show that the ability of an adversary to predict whether a given sample is memorized or not lower-bounds the advantage of any adversary in predicting the membership of that sample (§ 4). Lastly, we propose a practical instantiation of a highly effective MI strategy on memorized samples that can be performed with existing attacks. Extensive evaluation on image recognition models using four popular MI attacks (Yeom et al., 2018; Shokri et al., 2017; Song & Mittal, 2021; Carlini et al., 2022a) confirm that the susceptibility of MI risk is maximum, at times 100% (Advantage = 1) for memorized samples (§ 5).

## 2 BACKGROUND & RELATED WORK

**Notation.** The focus of our discussion is on supervised machine learning (ML). Here, one wishes to learn a trained model using data-points of the form $z = (x, y)$, where $x \in \mathcal{X}$ is the space of inputs and $y \in \mathcal{Y}$ is the space of outputs. A distribution $\mathcal{D}$ captures the space of inputs and outputs, and a dataset $D$ of size $T$ is sampled from it (*i.e.,* $D \sim \mathcal{D}^T$) such that $D = \{z_i\}_{i=1}^{T}$. Using this dataset, a learning algorithm $L$ can create a trained model $\theta \sim L(D)$ by minimizing a suitable objective $\mathcal{L}$.

### 2.1 MEMBERSHIP INFERENCE ATTACKS

Membership inference attacks (MIAs) are training distribution-aware, and aim to infer if a particular sample $z$ was present in the training dataset given access to a model $\theta$. One common theory for why such an attack exists relies on the belief that models *overfit* to their training data (Yeom et al., 2018), leading to numerous follow-up works aiming to infer the advantage of MIAs (in determining membership) (Mahloujifar et al., 2022; Humphries et al., 2020; Erlingsson et al., 2019; Jayaraman et al., 2020; Thudi et al., 2022). Prior MIAs approach this problem by training *shadow models* (*i.e.,* models trained on different subsets of the training data) and observing their outputs to infer membership outcomes (Shokri et al., 2017). Recent work by Carlini et al. (2022a) employs a similar approach, and combines the shadow model training with hypothesis testing (also performed in prior work (Murakonda et al., 2021; Ye et al., 2022)) to yield success. Their work, however, remarks that attacks are most representative at low false positive rate (FPR) regimes. In our work, we will use the MIA of Carlini et al. (2022a); more details are presented in Appendix B.1. Other approaches towards estimating MI success involve measuring entropy (Song & Mittal, 2021).

### 2.2 OOD DETECTION

The goal of out-of-distribution (OOD) detection is to determine if an input is drawn from the same distribution as the training dataset (*i.e.,* in-distribution) or not. In empirical settings, OOD samples are realized as data samples from different datasets (usually with disjoint label sets) (Nguyen et al., 2015; Ming et al., 2022). Most prior work formalizes the detection problem by designing a scoring function and applying a threshold. Performance is measured in terms of the false positive rate (FPR95) of OOD examples at the threshold when true positive rate (TPR) of in-distribution examples is as high as 95%, along with AUROC and AUPR. Starting with a baseline for OOD detection which uses maximum softmax probability (MSP) (Hendrycks & Gimpel, 2016), recent studies have designed various scoring functions based on the outputs from the final or penultimate layer (Liu et al., 2020; DeVries & Taylor, 2018), or a combination of different intermediate layers of DNN model (Lee et al., 2018; Raghuram et al., 2021).

### 2.3 LABEL MEMORIZATION

Label memorization, as noted by Feldman & Zhang (2020), is a concept where ML models remember information from training samples (likely) entirely to ensure good performance (when evaluated on those or similar samples). Brown et al. (2021) note that for certain types of inputs (which we shall define soon), memorization is essential to create a performant model. We begin by defining certain salient concepts related to memorization as in their original work.

1. **Label Memorization (Feldman & Zhang, 2020):** For a training algorithm $L$ on a dataset $D = \{z_i\}_{i=1}^T$ where $z_i = (x_i, y_i)$, the amount of label memorization by $L$ for any $z_i \in D$ is

$$\texttt{mem}(L, D, i) := \text{Pr}_{\theta \sim L(D)}[\theta(x_i) = y_i] - \text{Pr}_{\theta \sim L(D^{\backslash i})}[\theta(x_i) = y_i]$$

Since estimating the aforementioned quality is computationally prohibitive, Feldman & Zhang (2020) utilize an estimator based on sampling (refer Lemma 2.1 in their work). Such sampling-based estimators are also used in prior work related to influence (Ilyas et al., 2022; Kwon et al., 2021).

2. **Singletons:** As noted by Feldman (2020), datasets can be viewed as long-tailed mixtures of sub-populations. Models learnt from such datasets are unable to predict accurately on a sub-population until they are trained at least on one (representative) sample from the sub-population. Such a sample is called a *singleton*. Singletons may come from rare (or outlier) sub-populations or atypical sub-populations; in the former – there is no accuracy impact, while the latter may significantly impact accuracy. To better illustrate the difference between rare and atypical, consider a dataset formed by the following equation: $y = 2x + \epsilon$ where (a) $y$ is quantized to the closest element in $\mathbb{N}$, (b) $\epsilon$ is noise, and (c) is a normally distributed element in $[0, 1]$. Those values of $x$ which lie in the tail of the distribution are atypical, but values of $x \gg 1$ are outliers. However, models are unable to discriminate between outlier and atypical (*i.e.,* of frequency $\frac{1}{T}$) sub-populations; they memorize the singletons regardless. Brown et al. (2021) note that if a dataset $D$ contains $n < T$ singletons, each represented by $d$ bits, a performant model (*i.e.,* one whose error is close to optimal) learnt using the dataset will memorize $\geq \Omega(n \cdot d)$ bits. We utilize this insight in motivating our claims.

## 3 MOTIVATION: NOT ALL SAMPLES ARE EQUAL

Our work is motivated by observations by Kulynych et al. (2019) and Carlini et al. (2022a) (which we recreate in Appendix D.3) who notice that the efficacy of MIAs is different for different sub-groups of data (within a dataset). While prior work notes that MI efficacy is higher for those samples that are OOD, we wish to verify if there are other characteristics of a dataset that may result in this behavior.

**Experimental Setup:** We visualize this effect by considering the following setting: we consider 2 different datasets (say $D_O^{tr}$ and $D_U^{tr}$) and mix them using a mixing ratio (*e.g.,* to obtain $D^{tr} = \alpha \cdot D_O^{tr} + (1 - \alpha) \cdot D_U^{tr}$ for different values of $\alpha \in [0, 1]$ [1]). In such settings, one dataset is an *over-represented population* (*i.e.,* $D_O^{tr}$ when the mixing co-efficient is greater than 0.5), and the other is an *under-represented population* (*i.e.,* $D_U^{tr}$). Note that $D_O^{tr}$ and $D_U^{tr}$ share the common set of labels. In our experiment, the over-represented population is fixed as entire train set of MNIST (LeCun et al., 2010) (60,000 labeled samples), while the choice of under-represented population varies. We consider three variants: (a) MNIST augmented by Hendrycks et al. (2022) (henceforth referred to as augMNIST), (b) cropped SVHN (Netzer et al., 2011), and (c) CIFAR-10 (Krizhevsky, 2009). We choose these variants based on their *distance* from MNIST (Alvarez-Melis & Fusi, 2020); larger the distance, more atypical is the variant w.r.t the chosen over-represented population.

To construct $D_U^{tr}$ (for a given variant), we choose a subset of samples (more details follow) from the corresponding training set; care is taken to ensure the consistent label assignment to both $D_O^{tr}$ and $D_U^{tr}$ [2]. All samples are (a) resized to match the dimension of MNIST samples ($28 \times 28 \times 1$), and (b) converted to grayscale[3]. We adopt CNN models from the work of Carlini et al. (2022a) (details in Appendix A.1) and ensure that the models are trained to exhibit good accuracy on both populations.

### 3.1 CONNECTIONS TO OOD DETECTION

As stated earlier, we wish to better understand the implication of the "nature" of a sample on its susceptibility to MI. Similar to Carlini et al. (2022a), we hypothesize that outlier (*i.e.,* OOD) samples are more susceptible. To this end, we choose a random subset of samples from each of the aforementioned variants as our under-represented population ($D_U^{tr}$) and train models as described

---

[1]That is, $\alpha = \frac{|D_O^{tr}|}{|D_O^{tr}| + |D_U^{tr}|}$

[2]Consistent and balanced label assignment across classes; *i.e.,* all "airplane" images in CIFAR-10 are labeled as class 0 along with digit "0" images of MNIST

[3]Gray Image = $0.2989 \times R + 0.5870 \times G + 0.1140 \times B$.

earlier. To achieve reasonable training and test error, we need at least 1000 samples for augmented MNIST and cropped SVHN, and at least 6000 CIFAR-10 samples (10% of the total dataset); all these samples are chosen such that there are equal number of samples for each class (details in Table 1).

In all the experiments, we compute the privacy score proposed by Carlini et al. (2022a) which captures the vulnerability of a sample against MIA (higher is more vulnerable). We then plot the privacy score against the score returned by the two representative OOD detectors by Hendrycks & Gimpel (2016) and Liu et al. (2020) (higher is more likely to be OOD) to measure the correlation between the two.

| Over-represented | Under-represented | $\alpha$ | # Samples | Distance ($\times 10^3$) | Model | Train Acc. (%) | | Test Acc. (%) | |
|---|---|---|---|---|---|---|---|---|---|
| | | | | | | $D_O^{tr}$ | $D_U^{tr}$ | $D_O^{te}$ | $D_U^{te}$ |
| MNIST | augmented MNIST | 0.99 | 1000 | 1.26 | CNN32 | $99.15 \pm 0.13$ | $89.10 \pm 1.7$ | $98.43 \pm 0.19$ | $65.88 \pm 1.39$ |
| | cropped SVHN | 0.99 | 1000 | 3.67 | CNN32 | $99.44 \pm 0.05$ | $90.07 \pm 0.66$ | $98.60 \pm 0.09$ | $64.76 \pm 1.31$ |
| | CIFAR-10 | 0.93 | 6000 | 3.27 | CNN64 | $99.27 \pm 0.86$ | $82.69 \pm 6.84$ | $98.18 \pm 0.47$ | $40.72 \pm 1.17$ |

Table 1: **Salient features of the datasets used in our experiments.** Distance represents the distance between $D_O^{tr}$ and $D_U^{tr}$ (larger is more). Accuracy reported in 95% confidence interval – for each iteration in bootstrapping, we include 70% of $D^{tr}$ as the training set, and repeat 4000 times (as done in Feldman & Zhang (2020)). We also ensure that each sample in $D^{tr}$ appears in the training set for half of the 4000 iterations, following the setup of Carlini et al. (2022a).

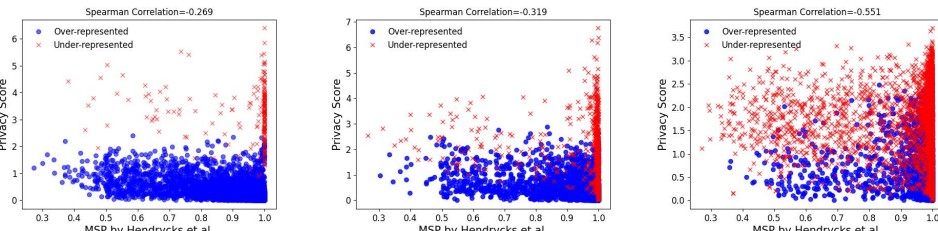

Figure 1: **Correlation between privacy score and OOD score.** (a) **Left**: CNN32 trained on MNIST+augMNIST (b) **Mid**: CNN32 trained on MNIST+SVHN (c) **Right**: CNN64 trained on MNIST+CIFAR-10.

**Results:** We capture correlations using the Spearman's rank correlation coefficient with $p$-value less than $0.001$ for statistical significance; absolute values less than $0.4$ are considered to represent a weak correlation, absolute values between $0.4$ and $0.59$ are considered to represent a moderate correlation, and absolute values greater than $0.6$ are considered to represent a strong correlation; the sign dictates whether the correlation is positive or not (which is not very useful for our discussion). From Figure 1, we observe that there is *no strong correlation* between OOD scores (measured by the approach of Hendrycks & Gimpel (2016)) and MI success; experiments with other OOD detector (Liu et al., 2020) show similar trends and are presented in Appendix D.2. This also suggests that one can not definitively use OOD scores to detect if a sample is a member or not. While some of the points with high privacy score (*i.e.,* highly susceptible to MI) are OOD points, others are not (this is particularly true in the case of Figure 1(c)). This suggests while the observations in prior work are true (to some degree), they are rather coarse and more refinement is needed.

> There is only a weak correlation between samples that are OOD (as detected by an OOD detector) and those that are very susceptible to MIAs.

## 3.2 CONNECTIONS TO (LABEL) MEMORIZATION

From the earlier results, we can see that not *all* points which are OOD are highly susceptible to MI. We wish to understand properties of those points that are susceptible. To this end, we borrow insight from the memorization literature, and identify those samples [4] (from the under-represented

---

[4] For the complete description of estimating privacy score and memorization degree, see Algorithm 1 in Appendix B.1.

population) that have high label memorization values ($> 0.8$, calculated based on the definition in § 2.3[5]). We then measure the correlation between the memorization value and the privacy score. Note that by doing so, we reduce the size of the under-represented population (details in Table 2).

| Over-represented | Under-represented | # Samples | | Model | Train Acc. (%) | | Test Acc. (%) | |
|---|---|---|---|---|---|---|---|---|
| | | $|D_O^{tr}|$ | $|D_U^{tr}|$ | | $D_O^{tr}$ | $D_U^{tr}$ | $D_O^{te}$ | $D_U^{te}$ |
| MNIST | augmented MNIST | 60,000 | 219 | CNN32 | $99.27 \pm 0.14$ | $72.08 \pm 5.93$ | $98.44 \pm 0.20$ | $49.38 \pm 3.04$ |
| | cropped SVHN | 60,000 | 209 | CNN32 | $99.32 \pm 0.10$ | $71.40 \pm 6.22$ | $98.48 \pm 0.20$ | $19.67 \pm 2.82$ |
| | CIFAR-10 | 60,000 | 2153 | CNN64 | $99.43 \pm 0.35$ | $72.66 \pm 1.86$ | $98.28 \pm 0.31$ | $20.78 \pm 1.215$ |

Table 2: **Salient features of the datasets used in our experiments.** Accuracy reported in $95\%$ confidence interval over 4000 iterations as in Table 1.

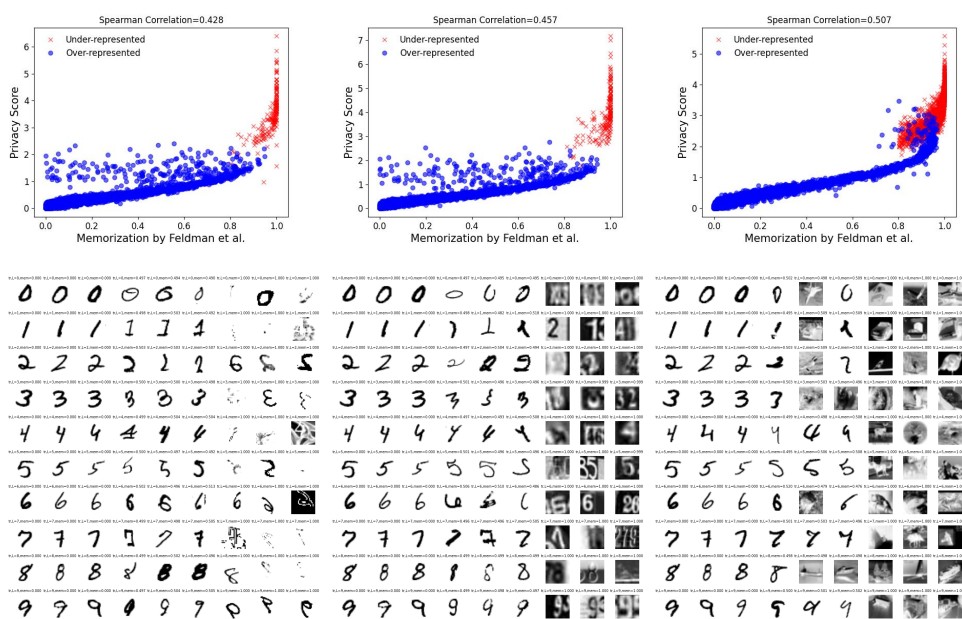

Figure 2: (a) **Left**: CNN32 trained on MNIST and augmented MNIST (b) **Mid**: CNN32 trained on MNIST and SVHN (c) **Right**: CNN64 trained on MNIST+CIFAR-10; (i) **Top**: Privacy score vs. memorization, (ii) **Bottom**: Random examples whose memorization value is close to 0 (first three columns), 0.5 (next three columns), 1 (last three columns). Each row shows images belonging to each of 10 classes.

**Results:** From Table 2, observe that the train accuracy of the samples from the under-represented population is reasonable, and the test accuracy associated to points from that class is better than random guessing (but not by much); this suggests that memorization has happened to some degree (Feldman & Zhang, 2020; Brown et al., 2021). Several observations can be made from Figure 2. First, the samples that have the high memorization values (*i.e.,* the red points) roughly correspond to those samples which have a high privacy score in Figure 1; this suggests that samples with high memorization values are indeed those that are highly susceptible to MIAs. Next, we visualize the examples of memorization values from each dataset (see the bottom of Figure 2). It confirms our intuition; the samples from the over-represented population fall into the low-memorization regime, atypical samples from the over-represented population (and a few similar samples from the under-represented population) compose the mid-level memorization regime, and the under-represented samples belong to the high-memorization regime. Finally, we observe that the correlation in this case is better; this suggests that memorization captures better signals regarding susceptibility to MIAs.

---

[5]We observe the privacy onion effect (Carlini et al., 2022b) when we recalculate memorization values on the chosen subset. More details are presented in Appendix D.2.

> **Remark**: Highly memorized samples (*i.e.,* singletons) are also highly susceptible to MIAs.

Thus, based on the experiments we have conducted, we conclude that there are connections between label memorization and MI susceptibility. In the next section, we formalize this connection through the form of a MI game, and calculate the adversary's advantage of success.

## 4 FORMALIZING THE CONNECTION

In § 3, we see that the memorized samples are also very susceptible to MI. In this section, we formally establish the connection between (high values of) memorization and MI success. We begin by formalizing a new MI game in § 4.1, and describe instantiations of the game that are both ineffective (§ 4.2) and effective (§ 4.3).

### 4.1 THE MI GAME

We extend the formulation of Mahloujifar et al. (2022) to introduce a game called $\mathtt{MI}(L, A, m)$ between the challenger (which runs a learning algorithm $L$) and an adversary $A$, where $m$ is a parameter which controls how many additional samples $A$ can use. Note that this a generalization of games presented in the literature before; roughly speaking, previous games correspond to $m = 1$.

**Step 1.** The adversary $A$ picks a dataset $D = \{z_1, \cdots, z_T\}$ and $D_A = \{z'_1, \cdots, z'_m\}$.

**Step 2.** The challenger samples a bit $b$ uniformly at random and creates

$$D' = \begin{Bmatrix} D \cup D_A & \text{for } b = 1 \\ D & \text{for } b = 0 \end{Bmatrix}$$

**Step 3.** The challenger learns a model $\theta$ by running $L(D')$ and sends $\theta$ to the adversary.

**Step 4.** The adversary guesses a bit $b'$ and wins if $b' = b$.

The output of the game $\mathtt{MI}(L, A, m) = 1$ iff the adversary $A$ wins. The advantage of adversary $A$ is defined to be $\mathtt{Adv}(L, A, m) = 2\Pr[\mathtt{MI}(L, A, m) = 1] - 1$, which can also be written as $2\Pr[b = b'] - 1$. Given a class of adversaries $\mathcal{C}$, one can define $\mathtt{Adv}(L, m)$ as $\sup_{A \in \mathcal{C}} \mathtt{Adv}(L, A, m)$. The following lemma holds in such cases.

**Lemma:** $\mathtt{Adv}(L, 1) \geq \frac{1}{m} \cdot \mathtt{Adv}(L, m)$

**Proof:** This result follows by a simple hybrid argument. Essentially the game $\mathtt{MI}(L, A, m)$ can be viewed as a series of $m$ games where the adversary only adds one additional point.

### 4.2 A SIMPLE (YET INEFFECTIVE) ATTACK

Consider the game described above, but for $m = 1$. Here, $D_A = \{z'\}$, where $z' = (x, y)$ is an arbitrary sample. The adversary $A'$ utilizes the following strategy to determine (non-)membership:

$$b' = \begin{Bmatrix} 1 & \text{if } \theta(x) = y \\ 0 & \text{if } \theta(x) \neq y \end{Bmatrix}$$

Such a strategy is not uncommon; it has been evaluated by (Yeom et al., 2018) and (Song & Mittal, 2021) and shown to be ineffective. In such a setting, it is easy to show how $\mathtt{Adv}(L, A', 1) = \mathtt{mem}(L, D, z')$ (refer Appendix C for the full derivation). However, note that this connection between memorization and MI is for (a) a particular MI strategy (as defined above) and not all (or even the best) of them, and (b) for any arbitrary sample and not the highly memorized samples. In the subsections that follows, we bridge this gap.

### 4.3 More Effective Adversaries

Consider the game described above, but where the adversary chooses $D_A$ s.t. it has $m$ points which are *all singletons* through interaction with some memorization oracle $\mathcal{O}_{\text{mem}}$. We modify the game this way for two reasons.

**Reason 1:** We define the empirical risk over a dataset $D'$ is defined as $R_{D'}(\theta) = \mathbb{E}_{(x,y)\in D'}[\mathbb{I}(\theta(x) = y)]$. As noted by Feldman (2020)[6], $R_{D_A}(\theta = L(D)) < R_{D_A}(\theta = L(D \cup D_A))$ for $D_A$ (comprised of singletons). We refer to this as *the memorization-induced risk gap* (MIRG). Note that such a gap is more prevalent when $D_A$ contains $m > 1$ points (as captured in our game definition). Thus, an adversary $A_{acc}$ can utilize this information, and its advantage is calculated as follows:

$$\texttt{Adv}(L, A_{acc}, m) = \Pr[R_{D_A}(\theta = L(D \cup D_A)) < \gamma] - \Pr[R_{D_A}(\theta = L(D)) < \gamma]$$

for some pre-determined risk threshold $\gamma$.

**Reason 2:** As noted in § 2.3, for inputs of size $d$, well-generalized models (*i.e.,* ones with low error) memorize $\Omega(m \cdot d)$ bits. It is clear to see for any adversary $A_{mem}$ that uses this information,

$$\texttt{Adv}(L, A_{mem}, m) = \Pr_{\theta \sim L(D \cup D_A)}[\theta(x) = y] - \Pr_{\theta \sim L(D)}[\theta(x) = y]$$

Both the aforementioned adversaries utilize (a) the fact that $D_A$ was chosen in such a manner so as to (b) exploit specific properties associated with memorization. Let us generalize the findings from above. For ease of exposition, let us consider the scenario where $m = 1$ (and $D_A = \{z'\}$). In such a setting, the aforementioned adversaries can be abstracted by an adversarial strategy $\texttt{Strat}$ which when given access to a model $\theta$ and singleton $z'$ returns the probability of the singleton being memorized by the model. Such a strategy is also a particular instantiation of an MIA (*i.e.,* if the probability is higher than some threshold, the sample is memorized and is consequently a member), and is a lower bound for $\sup_{A \in \mathcal{C}} \texttt{Adv}(L, A, 1)$. Thus,

$$\texttt{Strat}(\theta, z') \leq \texttt{Adv}(L, 1)$$

> **Conjecture:** If $z'$ is a singleton and $\theta$ is a well generalized model, there always exists a strategy $\texttt{Strat}(\theta, z') \to 1$.

The consequence of the above conjecture is that if there exists an effective test for checking *if* a singleton is memorized, then it can be used for a highly successful MIA. We use the term "if" because the mutual information bounds presented in the work of Brown et al. (2021) are a lower bound of the number of bits that are actually memorized.

We believe the aforementioned conjecture is true because both $A_{mem}$ and $A_{acc}$ are abstractions for $\texttt{Strat}$ (as noted earlier). When MIRG is true, then there exists a $\gamma$ such that $\texttt{Adv}(L, A_{acc}, m) = \Pr[R_{D_A}(\theta = L(D \cup D_A)) < \gamma] - \Pr[R_{D_A}(\theta = L(D)) < \gamma] \to 1$. Similarly, if $D_A$ is fully memorized, then $\Pr_{\theta \sim L(D \cup D_A)}[\theta(x) = y] - \Pr_{\theta \sim L(D)}[\theta(x) = y] \to 1$. We present more evidence in Appendix D.1. Formally proving the aforementioned conjecture remains an *open question*.

However, to ensure practicality, two considerations remain. First, how does one instantiate $\mathcal{O}_{\text{mem}}$ (needed for choosing $D_A$)? Effectively designing such an oracle is crucial for the success of a memorization-aware MIA. Next, and more importantly, how does one design a test for memorization (*i.e.,* instantiate the best $\texttt{Strat}$)? Prior work related to label memorization aims to estimate the propensity with which a sample is likely to be memorized by a model during the training process. However, we require a test that tells us *the likelihood of* a particular sample being memorized in the parameters of a trained model. In the section that follows, we answer these questions.

## 5 MI Attacks on Highly Memorized Samples

Having discussed the connections between memorization and MI advantage, we wish to understand if they hold in practice. In this section, we carry out experiments to address the following question: *are singletons highly vulnerable to MIAs?* In § 5.1, we describe the setup of our experiments. In § 5.2, we present results that affirmatively answers the aforementioned question.

---

[6]Refer Section 2.5 and Corollary 4.6 in (Feldman, 2020).

## 5.1 EXPERIMENTAL SETUP

**Goal.** We wish to understand if the efficacy of MIAs is higher on (a) OOD samples which are also singletons (*i.e.,* samples which are highly likely to be memorized) compared to (b) a mixture of different data-points (*i.e.,* singletons and samples from an over-represented population). We want to compare (a) with (c) an under-represented population filled with randomly chosen OOD samples.

**MIAs.** We consider four representative attacks from literature (Yeom et al., 2018; Shokri et al., 2017; Song & Mittal, 2021; Carlini et al., 2022a) (see Appendix A.1 for parameters associated with each attack). As noted in § 2.1, Carlini et al. (2022a) suggest that more emphasis should be placed on whether an adversary can *reliably* violate the privacy of even a few samples; this is captured through the TPR at low FPR. Following this idea, for each attack, we report TPR when a decision threshold $\tau$ is chosen to maximize TPR at a target FPR (*e.g.,* as low as $0.1\%$ in our setting) along with AUROC values (*i.e.,* the probability that a positive example would have higher value than a negative one).

**Datasets and Models.** Similar to our setup in § 4, we evaluate MIAs against ML models trained on dataset $D^{\text{tr}}$ that is a mixture of two different populations. We set $D_{\text{O}}^{\text{tr}}$ as MNIST (60,000 samples) throughout the experiment. We then consider two ways for an adversary to instantiate $\mathcal{O}_{mem}$ to obtain $D_{\text{U}}^{\text{tr}}$: (a) choosing random samples which are OOD (*e.g.,* 1000 MNIST samples augmented by Hendrycks et al. (2022), 1000 SVHN samples, or 6000 CIFAR-10 samples) as in § 3.1, or (b) using samples that are OOD but also highly memorized (*i.e.,* singletons whose memorization degree are approximated by the approach of Feldman & Zhang (2020), and are above $0.8$). As done earlier, care was taken to ensure consistent labeling. We use the same models as earlier, and ensure that the accuracy (both test and train) on both the under and over-represented populations is reasonable.

| Method | Dataset ($D^{\text{tr}}$) | AUROC $\uparrow$ | | TPR @ $0.1\%$ FPR $\uparrow$ | |
|---|---|---|---|---|---|
| | | All | Under-represented | All | Under-represented |
| Yeom et al. (2018) | MNIST+augMNIST (random) | 0.51 | 0.65 | 0.0% | 0.0% |
| | MNIST+augMNIST (singletons) | 0.50 | **0.91** | 0.0% | 0.0% |
| | MNIST+SVHN (random) | 0.50 | 0.74 | 0.0% | 0.0% |
| | MNIST+SVHN (singletons) | 0.50 | **0.95** | 0.0% | **2.61%** |
| | MNIST+CIFAR-10 (random) | 0.51 | 0.81 | 0.0% | 0.0% |
| | MNIST+CIFAR-10 (singletons) | 0.50 | **0.96** | 0.0% | 0.0% |
| Shokri et al. (2017) | MNIST+augMNIST (random) | 0.51 | 0.72 | 0.20% | 0.70% |
| | MNIST+augMNIST (singletons) | 0.51 | **0.99** | 0.15% | **18.88%** |
| | MNIST+SVHN (random) | 0.52 | 0.78 | 0.19% | 3.01% |
| | MNIST+SVHN (singletons) | 0.51 | **0.98** | 0.14% | **15.69%** |
| | MNIST+CIFAR-10 (random) | 0.58 | 0.89 | 0.82% | 2.49% |
| | MNIST+CIFAR-10 (singletons) | 0.55 | **0.99** | 1.92% | **17.21%** |
| Song & Mittal (2021) | MNIST+augMNIST (random) | 0.51 | 0.70 | 0.16% | 0.53% |
| | MNIST+augMNIST (singletons) | 0.51 | **0.98** | 0.10% | **16.12%** |
| | MNIST+SVHN (random) | 0.50 | 0.77 | 0.17% | 2.86% |
| | MNIST+SVHN (singletons) | 0.51 | **0.98** | 2.04% | **14.11%** |
| | MNIST+CIFAR-10 (random) | 0.54 | 0.87 | 0.20% | 2.06% |
| | MNIST+CIFAR-10 (singletons) | 0.53 | **0.99** | 1.98% | **15.53%** |
| Carlini et al. (2022a) | MNIST+augMNIST (random) | 0.53 | 0.87 | 1.45% | 27.55% |
| | MNIST+augMNIST (singletons) | 0.53 | **1.0** | 1.07% | **100%** |
| | MNIST+SVHN (random) | 0.53 | 0.95 | 2.15% | 51.5% |
| | MNIST+SVHN (singletons) | 0.53 | **1.0** | 1.24% | **100%** |
| | MNIST+CIFAR-10 (random) | 0.59 | 0.96 | 5.7% | 44.2% |
| | MNIST+CIFAR-10 (singletons) | 0.56 | **1.0** | 4.6% | **96.5%** |

Table 3: **Comparison of representative MIAs.** For each attack, we evalutate the performance on three different mixture of datasets: MNIST with (a) augmented MNIST generated by the technique of Hendrycks et al. (2022), (b) SVHN, and (c) CIFAR-10. For each choice of MIA and mixture dataset, we direct readers to compare AUROC and TPR between (a) (All) vs. (Under-represented), and (b) (random) vs. (singletons) to see the effect of utilizing singletons in MIAs. Best results are **boldfaced**.

## 5.2 RESULTS

The results of our experiments are presented in Table 3. Overall, the attack of Carlini et al. (2022a) outperforms other attacks in the low FPR regime ($0.1\%$), as illustrated in their paper (also explaining the overall poor performance of the attack of Yeom et al. (2018)).

We observe that every attack performs better with respect to the samples in under-represented population, which are drawn from different population than the over-represented population (*i.e.,* MNIST samples). For instance, given MNIST+SVHN (random), the attack of Shokri et al. (2017) (the earliest known technique) yields an overall AUROC of $0.52$ and TPR of $0.19\%$ when evaluated on all samples. However, we see an increase in the AUROC to $0.78$ and TPR to $3.01\%$ when the attack is evaluated on under-represented SVHN samples.

The same observation applies when attacks are tested on highly memorized OOD samples. Consider the case of MNIST+CIFAR-10 (singletons): by utilizing the highly memorized samples, the attack of Song & Mittal (2021) can improve its efficacy from AUROC=0.53 and TPR=1.98% to AUROC=0.99 and TPR=15.53%. This is even higher than when the same attack is evaluated on just (random) OOD samples (*i.e.,* AUROC=0.87 and TPR=2.06%).

It is noteworthy that the attack of Carlini et al. (2022a) achieves near-perfect AUROC and TPR even at the low-FPR regime on the under-represented population comprising of highly memorized OOD samples in any of three dataset mixture settings. We believe this high accuracy is attributed to the commonalities between the attack and memorization estimation (refer Appendix B.2) *i.e.,* the attack is estimating memorization as well.

> **Result:** With the knowledge of memorization, performance of MIAs improves significantly.

## 6 DISCUSSION

**1. The Privacy Onion Effect:** The definition of label memorization is implicitly dependent on the dataset (refer § 2.3). In recent work, Carlini et al. (2022b) note that if a set of highly memorized points is removed from a dataset, points which previously had low memorization values now have high values (and vice versa). They term this the *privacy onion effect* (and we observe the same in Figure 5 (bottom row) in Appendix D.2. While defenders against memorization-guided MIAs may consider removing those samples that are likely to be memorized from the training dataset, two problems may emerge. The first is that the error of the model (on the sub-population associated with the deleted singleton) shoots up. Secondly, and more importantly, there will be a new set of points which are highly likely to be memorized. This suggests that attackers will always win.

**2. Connections to Privacy Auditing:** When training models with differential privacy (DP) (Abadi et al., 2016), the estimate for the privacy budget ($\varepsilon$) obtained is very loose. To obtain tighter estimates, a rich body of literature (termed privacy auditing) aims at empirically estimating $\varepsilon$ (Nasr et al., 2021; Zanella-Béguelin et al., 2022; Jagielski et al., 2020). The core technique used in most of these works is to calculate an estimate of $\varepsilon$ using the DP-trained models susceptibility to highly-effective MIAs (which are typically designed for any arbitrary data-point). We believe that better estimates can be obtained when auditing is performed using memorization-guided MIAs. A key consideration is to understand the implications of DP-training on memorization of data-points.

**3. Data Reconstruction:** Brown et al. (2021) discuss approaches to reconstruct data from memorized models. For a given sample $z'$ and a trained model $\theta$, such approaches may also be used to instantiate $\texttt{Strat}(z', \theta)$; if reconstruction is successful, we can claim that the sample is memorized. Based on the results presented in § 5 of (Brown et al., 2021), we can see that such a $\texttt{Strat}(z', \theta) \rightarrow 1$, providing more motivation for the correctness of our conjecture.

## 7 CONCLUSION

In this work, we attempt to explain the high efficacy of MIAs for specific data-points as a function of their susceptibility to be memorized. Our theoretical justification provides motivation to our claims, and notes that the efficacy of attacks on such samples can be very high (*i.e.,* almost always accurate). We also experimentally verify our claims, and the results are in accordance with our expectations. More analysis is required to (a) convert the claims associated with mutual information from Brown et al. (2021) to probability bounds (needed to insantiate attack strategies), and (b) understand the efficacy of MI for samples that are unlikely to be memorized.

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

# Appendix

## A Implementation Details

We run all experiments with Tensorflow, Keras, Jax and NVDIA GeForce RTX 2080Ti GPUs.

### A.1 Experimental Setting.

**Datasets.**   Here we describe the datasets used throughout our paper.

1. **MNIST** (LeCun et al., 2010). The MNIST dataset consists of 60,000 handwritten digits (between 0 to 9) for train set, and 10,000 images for test set.

2. **augMNIST**. We generate the augmented variation of the MNIST dataset by using the technique in Hendrycks et al. (2022). Original MNIST image is mixed with augmented versions of itself and the grayscale version of provided 14248 fractal images. We use severity level of 2 for mixing, and severity level $= 4$ for base augmentation operations (such as normalization, random cropping, and rotation) for $4$ iterations for each image. We use 1,000 images randomly chosen from the augmented train set, while ensuring the balance across 10 classes.

3. **SVHN** (Netzer et al., 2011). SVHN is the color images of real world house numbers. We use the cropped version of the SVHN dataset, where images are tightly cropped around each digit of 0-9. From the original train/test set, we randomly select 1,000 images that are equally balanced across 10 labels.

4. **CIFAR-10** (Krizhevsky, 2009). We use randomly sampled 6,000 images from the original train set of 60,000 CIFAR-10 images. We preserve their original labels so that images for class $i$ of CIFAR-10 where $i \in [0, 9]$ are assigned to the same class as the MNIST images with label $i$.

**Models.**   We adopt CNN architectures from Carlini et al. (2022a): CNN models with 32 and 64 convolutional filters (referred to as CNN32 or CNN64, respectively). We use them when training shadow models or target models for MIAs.

**MIAs.**   We briefly describe the details of different membership inference attack methods.

1. Yeom et al. (2018). The attack is based on the observation that ML models are trained to minimize the loss of the training samples, and the loss values on them are more likely to be lower than the samples outside of the training set. They suggest applying threshold on the loss values from the ML model to infer the membership of an input.

2. Shokri et al. (2017). The attack uses a trained ML model to ascertain membership/non-membership. In our experiments, a fully-connected neural network with one hidden layer of size 64 with ReLU (rectifier linear units) activation functions and a SoftMax layer is used to distinguish feature vectors obtained from shadow models trained with and without a data-point.

3. Song & Mittal (2021). The attack uses shadow models to approximate the distributions of entropy values, instead of cross-entropy loss. Given a target model and sample, they conduct hypothesis test between the member and non-member distributions for each class.

4. Carlini et al. (2022a). Refer Algorithm 1 in Appendix B.1. We used the implementation provided by the authors for the attack.

## B Discussion on OOD Detection and MI

### B.1 Algorithm for MIA and Memorization

We describe the algorithm proposed by Carlini et al. (2022a) for MI (reproduced from their work), and Feldman & Zhang (2020) for label memorization below. Observe that both the MI success and

---

**Algorithm 1** Algorithm for estimating memorization and privacy score.

---

**Require:** model $f$, example $z = (x, y)$, data distribution $\mathcal{D}$
1: $\text{confs}_{\text{in}} = \{\}$
2: $\text{confs}_{\text{out}} = \{\}$
3: $\mathcal{F}_{\text{in}} = \{\}$
4: $\mathcal{F}_{\text{out}} = \{\}$

5: **for** $N$ times **do**
6:     $D_{\text{attack}} \leftarrow^{\$} \mathcal{D}$            ▷ *Sample a shadow dataset*
7:     $\theta_{\text{in}} \leftarrow L(D_{\text{attack}} \cup \{(x, y)\})$           ▷ *train IN model*
8:     $\mathcal{F}_{\text{in}} \leftarrow \mathcal{F}_{\text{in}} \cup \{\theta_{\text{in}}\}$
9:     $\theta_{\text{out}} \leftarrow L(D_{\text{attack}} \backslash \{(x, y)\})$          ▷ *train OUT model*
10:    $\mathcal{F}_{\text{out}} \leftarrow \mathcal{F}_{\text{out}} \cup \{\theta_{\text{out}}\}$
11:    $\text{confs}_{\text{in}} \leftarrow \text{confs}_{\text{in}} \cup \{\phi(\theta_{\text{in}}(x)_y)\}$
12:    $\text{confs}_{\text{out}} \leftarrow \text{confs}_{\text{out}} \cup \{\phi(\theta_{\text{out}}(x)_y)\}$
13: **end for**

14: $\mu_{\text{in}} \leftarrow \texttt{mean}(\text{confs}_{\text{in}})$
15: $\mu_{\text{out}} \leftarrow \texttt{mean}(\text{confs}_{\text{out}})$
16: $\sigma_{\text{in}}^2 \leftarrow \texttt{var}(\text{confs}_{\text{in}})$
17: $\sigma_{\text{out}}^2 \leftarrow \texttt{var}(\text{confs}_{\text{out}})$
18: $\widetilde{\text{mem}}(\mathcal{A}, S, z) := \Pr_{\theta_{\text{in}} \in \mathcal{F}_{\text{in}}}[\theta_{\text{in}}(x) = y] - \Pr_{\theta_{\text{out}} \in \mathcal{F}_{\text{out}}}[\theta_{\text{out}}(x) = y]$

19: **return** $\frac{|\mu_{\text{in}} - \mu_{\text{out}}|}{\sigma_{\text{in}} + \sigma_{\text{out}}}$ and $\widetilde{\text{mem}}(\mathcal{A}, S, z)$

---

label memorization can be obtained from the same algorithm by saving some additional state. As noted earlier, for a single sample, the algorithm requires $2 \cdot N$ training runs, making it prohibitively expensive for large values of $N$. For $T$ samples, it requires $2 \cdot T \cdot N$ training runs.

### B.2 COMMON SUBROUTINES

Notice that the privacy score estimation relies on training numerous models, some with the point under consideration, and some without (refer the grey box in Algorithm 1). We refer to this as the leave-one-out (LOO) subroutine.

Observe that the exact same LOO subroutine can be used to empirically estimate memorization (which is also used to measure algorithmic stability and influence). The attack by Carlini et al. (2022a) couples this with hypothesis testing, leading to high success rates for all samples. However, if we restrict our focus to those samples which are highly likely to be memorized, we can observe that the generalization error of models without these samples is not close to optimal (Feldman, 2020).

## C PROOF FOR APPROACH IN § 4.2

$$
\begin{aligned}
\texttt{Adv}(L, A, 1) &= 2 \cdot \Pr[b' = b] - 1 \\
&= 2 \cdot \{\Pr[b = 1] \cdot \Pr[b' = 1 | b = 1] + \Pr[b = 0] \cdot \Pr[b' = 0 | b = 0]\} - 1 \\
&= \Pr[b' = 1 | \theta \sim L(D \cup z')] + \Pr[b' = 0 | \theta \sim L(D)] - 1 \\
&= \Pr[b' = 1 | \theta \sim L(D \cup z')] + (1 - \Pr[b' = 1 | \theta \sim L(D)]) - 1 \\
&= \Pr_{\theta \sim L(D \cup z')}[\theta(x) = y] - \Pr_{\theta \sim L(D)}[\theta(x) = y]\} \\
&= \texttt{mem}(L, D, z' = (x, y))
\end{aligned}
$$

# D    ADDITIONAL RESULTS

## D.1    EVIDENCE: DISTRIBUTION OF SOFTMAX SCORES

A model $\theta$ returns a vector when fed an input $x$, where $\theta(x)_i$ corresponds to the $i^{th}$ element of the prediction vector. The maximum softmax score is defined as $\max_i \theta(x)_i$.

For the experiment that follows, we consider a dataset which is a mixture of MNIST (over-represented population) and singletons obtained from CIFAR-10 (under-represented population). In Figure 3, we plot $\beta = \max_i \theta_{in}(x)_i - \max_i \theta_{out}(x)_i$ where $\theta_{in}$ is obtained when the model is trained with the sample $(x, y)$ and $\theta_{out}$ is obtained when the model is trained without the sample $(x, y)$. Intuitively, $\beta$ will be close to 0 when the presence or absence of the model does not significantly impact the model's performance (*i.e.,* $\max_i \theta_{in}(x)_i = \max_i \theta_{out}(x)_i$). However, $\beta > 0$ when the sample's presence/absence influence performance (*i.e.,* $\max_i \theta_{in}(x)_i > \max_i \theta_{out}(x)_i$). Observe that for the under-represented population, $\beta > 0$. An adversary can exploit this information (*i.e.,* decrease in model's performance) to perform MI.

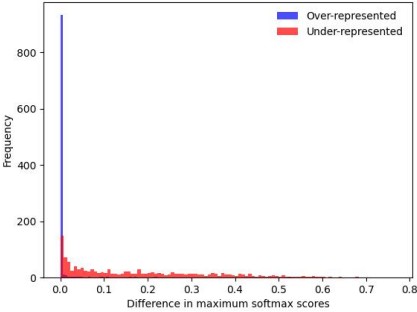

Figure 3: **Distribution of softmax score gap.** Observe that the score gap of the under-represented populations is larger than that of the over-represented populations.

## D.2    ADDITIONAL RESULTS FROM § 3

Figure 4 conveys the same information as Figure 1 but when the OOD-ness of a sample is calculated using the approach of Liu et al. (2020). Observe that there is low correlation between the OOD-ness of a sample and its susceptibility to MIAs.

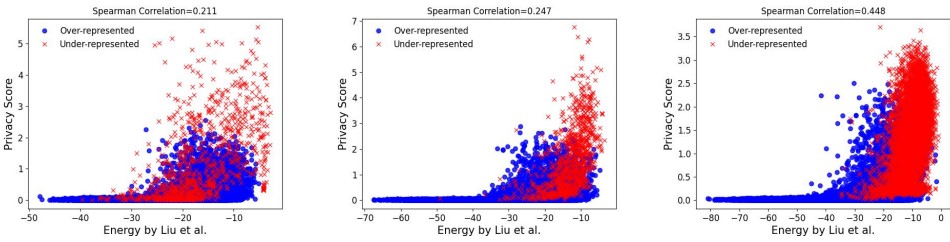

Figure 4: (a) **Left**: CNN32 trained on MNIST+augMNIST (b) **Mid**: CNN32 trained on MNIST+SVHN (c) **Right**: CNN64 trained on MNIST+CIFAR-10. For the OOD detector by Liu et al. (2020), higher the score, more likely is a sample to be OOD.

To obtain the red points in Figure 2, we first choose those samples from Figure 1 which are from the under-represented population and plot their memorization values (which are presented in the top row of Figure 5). Next, we consider those red points whose memorization value is greater than 0.8 (from the top row of Figure 5) and include them in a dataset of over-represented MNIST samples and re-calculate the memorization of those samples (to obtain the bottom row in Figure 5). Figure 2 is

obtained by choosing those red points whose memorization value is greater than 0.8 in the bottom row of Figure 5.

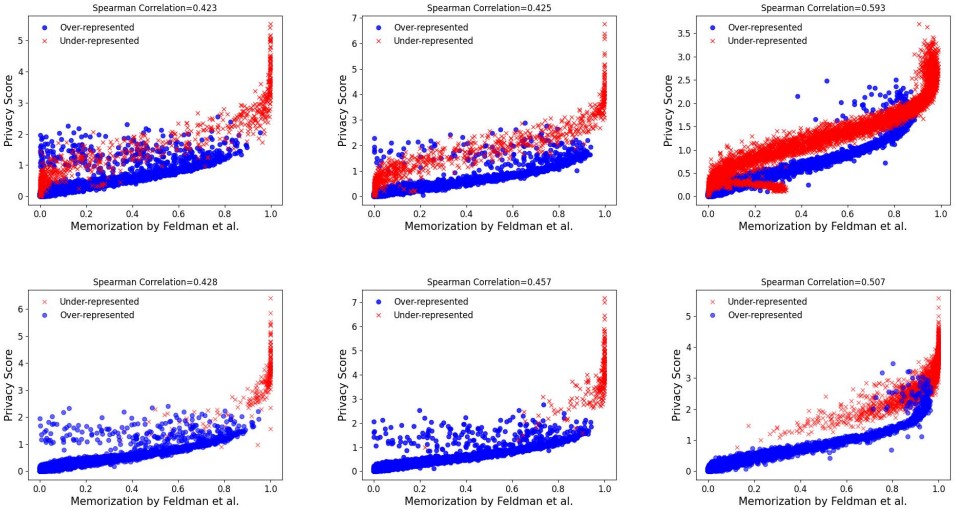

Figure 5: (a) **Left**: CNN32 trained on MNIST+augMNIST (b) **Mid**: CNN32 trained on MNIST+SVHN (c) **Right**: CNN64 trained on MNIST+CIFAR-10.
**Top Row:** Memorization values of under-represented samples vs. privacy score.
**Bottom Row:** Recalibrated memorization values of under-represented samples from the top row (whose original memorization value was greater than 0.8) vs. privacy score.

### D.3 Additional Results: Motivation

Figure 6 is our recreation of Figure 13 from Carlini et al. (2022a), which provides an observation that OOD samples are more susceptible to MI than others. It is obtained using a mixture of MNIST and CIFAR-10 (as the under-represented population). In Figure 6, we see that the privacy score is separated (into 2 normal distributions) when the data for the under-represented population is comprised of singletons. This distinction is not as clear when the under-represented population's data are not singletons.

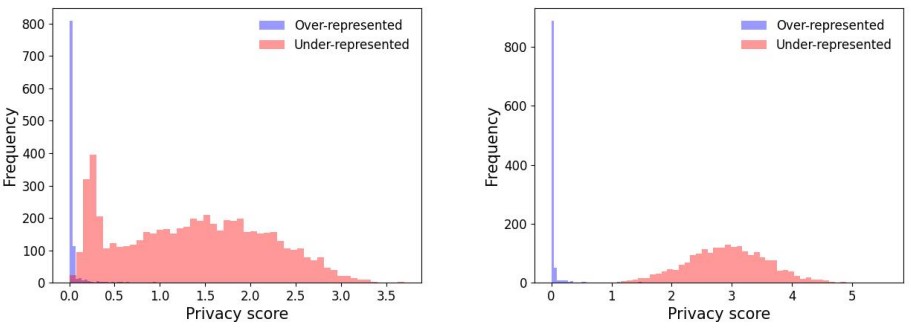

Figure 6: **Singletons are more effective than random OOD samples in enlarging the privacy score gap between data populations. Left**: MNIST+CIFAR-10 (random), **Right**: MNIST+CIFAR-10 (singleton)

