# OpenReview forum: "Exploring Connections Between Memorization And Membership Inference"
_ICLR.cc/2023/Conference — Submitted to ICLR 2023_

### Official Review · Reviewer_zop7 · 2022-10-23

**Confidence:** 3
**Correctness:** 3
**Technical Novelty And Significance:** 2
**Empirical Novelty And Significance:** 3
**Recommendation:** 6

**Clarity, Quality, Novelty And Reproducibility:**

The paper is well-written and easy to follow. The research topic is important and interesting.

**Strength And Weaknesses:**

Strengths:

1. The paper is written clearly and the studied topic is important and interesting.

2. The empirical results are strong enough (especially Fig. 1 and Fig. 2) to justify the correlation between label memorization and membership inference.


Concerns & Questions:

1. According to the definition of label memorization score, one needs to access the learning algorithm to calculate such a score. However, in real-world attack scenarios, the learning algorithm (as well as hyperparameters) of a black-box model may be agnostic by adversaries, which would result in difficulties in obtaining memorization scores. Please discuss the practicality of leveraging memorization scores in membership inference attacks.

2. Why the conjecture in Section 4.3 is appropriate? Please justify.

3. In Table 1, every model is trained with MNIST and a subset of under-represented data. However, from Table 2, it seems that every model is instead trained with MNIST and the singletons from the under-represented data (which results in the size of the under-represented population being reduced). I think the comparison between Table 1 and Table 2 is unfair, and a more fair comparison may be that: (1) train model on MNIST + subset of under-represented data, and (2) compare the classification accuracy of the model on random under-represented data and under-represented singletons. Please comment.

4. Only MNIST is used as the over-represented dataset. I suggest using more datasets as the over-represented datasets.

**Summary Of The Paper:**

This paper studies in what situation training samples are vulnerable to membership inference attacks.
Firstly, the authors empirically argue that membership inference performance may have a weak connection with the OOD property of training data.
Then, they hypothesize that membership inference advantages may be related to how well the samples are memorized by models.
Through experiments, the authors find a clear correlation between the label memorization score of data and the corresponding membership inference attack accuracy, which justifies their claim.

**Summary Of The Review:**

This paper studies an important and interesting problem, i.e., the relationship between data memorization and membership inference advantages. In general, the empirical results are strong enough to justify most of the claims in the paper. Therefore, I tend to accept this paper.

---

> ### Author Response · Authors · 2022-11-16
> **Thank you for your feedback!**
>
> Response to practicality: The reviewer makes an astute observation that in practical settings, the adversary will not have access to the learning algorithm. The objective of our work is not to show how one can launch a new MI attack, but rather how existing MI attacks are more effective for specific samples (i.e., those that are more likely to be memorized). Calculating the MI scores assumes knowledge of how the models are trained; the same assumptions are utilized in all MI attacks where the adversary requires training shadow models (based on knowledge of the training algorithms). This is assumed in the original work of Shokri et al. and is also true for the more recent work by Carlini et al. Cost-wise, learning the memorization scores is as expensive as the MI attack proposed by Carlini et al., as both utilize the same subroutines (observe the connection in Appendix B2 of our work).
>
> Response to appropriateness of conjecture in section 4.3: The conjecture states that, for a given data point and trained parameters, if there exists a mechanism to determine if these parameters memorize the data point, then that will serve as a lower bound for the advantage of an MI adversary in determining membership for that specific data point. We believe this is fairly intuitive (i.e., if an adversary can determine if a data point is likely to be memorized apriori, then it will likely be a member).
>
> Response to comparison b/w Tables 1 and 2: Could the reviewer kindly clarify the objective of this proposal? In the current setup, one can see that despite the (OOD) dataset size reduction in Table 2, there is only a small drop in the training accuracy of the model (but a drop in the test accuracy of the model). This validates that singletons introduced during training are useful only in predicting samples that are very similar to them (which may not be the case in a conventional test dataset). However, in the case of Table 1, we choose an OOD dataset to be large enough such that the model is (reasonably) well generalized (both in the context of training and testing) on said dataset. By doing so, we wanted to ensure that the model learns features from the OOD dataset, and this in-turn will be useful in launching effective OOD discrimination approaches and MI approaches (which hinge on the ability of the model to “fit well” to the sample, be it in-distribution or OOD).
>
> Response to using other datasets: Could the reviewer kindly clarify what they hope to see through this experiment? Running the experiments required for this submission involved extensive computation. Training classifiers needed for the Carlini et al. MI attack as well as classifiers needed for obtaining the memorization score required numerous training runs (note that the original papers were published by Google, where there is extensive compute capabilities available). Additionally, care must be taken to ensure that there is reasonable accuracy on the OOD datasets; this requires extensive hyperparameter tuning.
>
> We hope you find these answers satisfactory. If so, please do consider increasing the score for the submission. If not, kindly let us know what better we can clarify.

---

### Official Review · Reviewer_eP6o · 2022-10-24

**Confidence:** 4
**Correctness:** 2
**Technical Novelty And Significance:** 2
**Empirical Novelty And Significance:** 2
**Recommendation:** 3

**Clarity, Quality, Novelty And Reproducibility:**

The paper is generally well written, but understanding its main points is difficult because many important concepts are left underspecified.
For example, the concept of a "singleton", which is central to many of the paper's arguments, is never formally defined.

Similarly, the term "OOD" is used throughout the paper without ever being clear on what its formal definition actually is. As discussed below, there isn't any single canonical definition of OOD. Instead we just have different metrics that try to capture something like OOD. But it would be worth clarifying this early on in the paper, as the paper's results have a very different interpretation in this light.

Section 4.3 suffers from vagueness the most. The given conjecture uses terms such as "singleton", "well generalized" without any proper definition. The notation "-> 1" is also not defined. It is unclear what it would mean for Strat to "tend towards 1" (my interpretation of the meaning) as there are no variable quantities here. As a result, the conjecture is essentially meaningless as one cannot hope to confirm or refute it.

**Strength And Weaknesses:**

Strengths:
- understanding what makes data vulnerable to privacy attacks is an important problem
- comparing different ways of "defining" OOD data is valuable


Weaknesses:
- Very imprecise. Terms like "singleton" or "OOD" are never formally defined, and the conjecture in Section 4.3 and its consequences are stated in terms vague enough that the conjecture cannot meaningfully be falsified
- Circular reasoning:

**Summary Of The Paper:**

The paper considers the connection between label memorization and membership inference.
The paper shows that examples that are considered OOD according to some empirical metrics may not be more vulnerable to MI. Instead, examples with large label memorization (as defined by Feldman) are more likely to be vulnerable to MI attacks.

**Summary Of The Review:**

The paper's main claim is that OOD points are not more susceptible to MI, but this depends entirely how one defines "OOD". In fact, an entirely natural way of defining OOD data is simply by using the result of an MI attack.
What the paper actually shows (in Figure 1) is that some canonical OOD metrics (like MSP) are not aligned at all with privacy scores from MI attacks. But this can be interpreted in two ways: either privacy scores aren't impacted much by OOD data, or MSP is not actually a very good OOD metric!
Looking at Figure 1, it seems that the latter conclusion is most at "fault" here: indeed, in this experiment you have ground truth on what data is OOD (i.e., the under-represented data). As we can see, MSP is terrible at distinguishing the under-represented and the over-represented data. So it seems like a poor OOD metric.
In contrast, the privacy score appears to distinguish the under- and over- represented data a lot better.
There is a prior work by Carlini et al (https://arxiv.org/abs/1910.13427) that looks at the agreement between different measures of what constitutes an outlier, including privacy metrics. It would be worth discussing the relation of this work to that paper.

As figure 2 shows, the privacy score of Carlini et al and the memorization score of Feldman are more closely correlated. This is not particularly surprising though: from the definition of label memorization (which should be credited to Feldman 2020, rather than Feldman & Zhang), it is obvious that high memorization of a given point implies the ability to do membership inference on that point well.
In fact, the two metrics are essentially measuring the same thing, just at different levels of granularity: the score of Carlini et al. measures the distinguishability of the loss distributions when training on D vs D \ {i}. Label memorization takes a much coarser approach by just collapsing these distributions to a single scalar, the probability of outputting the correct label. Through this view, it is not clear why it should be surprising that these metrics are very strongly correlated.

Due to this very strong correlation, the results from section 5 also seem somewhat circular. Of course, if we focus the MI attacks on those samples where the label probabilities for members and non-members are most different, then the MI attack will work better.

So overall, it is not clear to me what conclusions to draw from this paper. There are some interesting results that seem to show that MI attacks are better at identifying OOD data compared to other metrics. But these results are not at all interpreted this way, and rather taken to imply that prior work somehow drew incorrect conclusions regarding the success of MI attacks on OOD data.

---

> ### Author Response · Authors · 2022-11-16
> **Thank you for your feedback!**
>
> Response to singleton definition: The term singleton is defined in section 2.3. While we agree that the definition may not be the most formal, we believe it captures (a) the essence of what we subsequently try to communicate, and (b) the definition as defined in prior work by Feldman and colleagues. We would be open to concrete suggestions on how the definition can be improved.
>
> Response to OOD definition: In the context of this work, we claim OOD-ness when there’s a mixture of two distributions. Concretely, one can visualize this as a mixture of two datasets. Could the reviewer please clarify what “these different metrics that try to capture something like OOD” that we have in our paper?
>
> Response to section 4.3: We apologize for not defining the arrow as “tends to”. We also apologize if the conjecture seems unclear to this reviewer. What we state is that, for a given data point and trained parameters, if there exists a mechanism to determine if these parameters memorize the data point, then that will serve as a lower bound for the advantage of an MI adversary in determining membership for that specific data point. We believe this is fairly intuitive (i.e., if an adversary can determine if a data point is likely to be memorized apriori, then it will likely be a member). We hope that this will allow the reviewer to think more about if it can be refuted.
>
> Response to goodness of OOD: Refer to the earlier comment on the definition of OOD. We would also like to point the reviewer to a more recent OOD detection method that we evaluated in Appendix D.2; the conclusions are similar to MSP. This suggests that the problem is not in the method, but the hypothesis itself.
>
> Response to new metrics: We will discuss relevant content from this arXiv paper; we would like to point out, however, that most of the analysis in this paper is empirical and not theoretically grounded.
>
> Response to relationship b/w works of Carlini & Feldman: We thank the reviewer for making this astute observation. We raise this observation in Appendix B.1 ourselves. However, the surprising fact is that the paper by Carlini et al. does not note this connection (despite the work being published after the work of Feldman and Zhang). In that regard, our work is the first to formally study the implication of this connection, bridging work that is published in two different communities (ML and Security). The observation being fairly intuitive is a testament to the presentation of results in our work. Because the two are deeply related, one can utilize knowledge of memorization to enhance the efficacy of MI for specific samples. This result is interesting in its own right because it is able to explain the “disparate efficacy of MI” through the lens of memorization; now we have a better understanding of the efficacy of MI for specific samples (relative to other samples that were used to train the model).
>
> Response re: conclusions: The reviewer makes an interesting observation that MI attacks are better than standard OOD approaches in identifying OOD data but that is not the primary conclusion we wish to highlight. The conclusion is that “for those samples that are at a higher risk of being memorized, they are more susceptible to MI”. With this conclusion, we are able to shed some light on the disparate performance of MI towards different samples through the lens of memorization.
>
> We hope you find these answers satisfactory. If so, please do consider increasing the score for the submission. If not, kindly let us know what better we can clarify.

---

### Official Review · Reviewer_j9KE · 2022-10-25

**Confidence:** 4
**Clarity, Quality, Novelty And Reproducibility:** This paper is clearly written and rel…
**Correctness:** 3
**Technical Novelty And Significance:** 2
**Empirical Novelty And Significance:** 2
**Recommendation:** 3

**Strength And Weaknesses:**

**Strength**: This paper systematically studied the relationship between memorization and MI attack performance, and also evaluated common OOD detection algorithms.

**Weakness**

1. The connection between label memorization and membership inference attack performance is known. The attack algorithm that this paper primarily use (Carlini et al. 2022a) is directly built on this relationship: it performs hypothesis testing based on the distribution of two group of shadow models to make the MIA prediction. Since the (practical estimator of the) label memorization is just the difference between the mean of the two distributions, the larger the difference between the two distributions, the easier the hypothesis testing is, and subsequently the higher the MIA performance is. So this observation, while not formally characterized yet, is essentially known in the literature.

2. This paper claims that OOD is not well correlated with MIA performance. But this is not well supported, and the main reason for the observation seems to be that the OOD detector used in the paper are not a good one. From Fig. 1, x-axis, we can see that the OOD score failed to make a clear separation between the over-represented data and under-represented (out of distribution) data.

3. The paper propose to perform MIA in a MI Game setting where the attacker choose the points being attacked according to the susceptability of them being memorized. This ended up making the MIA performance higher. I'm not an expert in security, but showing that the attackers could have high success rate when they choose to only attack the most outlier examples does not seem surprising or concerning, comparing to, e.g., showing that an attacker can achieve high rate on average across randomly sampled training examples.

**Summary Of The Paper:**

This paper studies the relationship between label memorization and membership inference. It showed that examples with high memorization are more easily attacked. Based on this, it formulated membership inference attack where the attacker choose highly memorized examples, and show that the attack performance improved significantly.

**Summary Of The Review:**

This paper studies the relationship between memorization and membership inference. My main concern over this paper is that the main empirical observations are essentially known, and the proposed MIA that focus on attacking the memorized examples is just a straightforward implication of the observation applied to existing MI attacks, and does not seem to bring new insights to the table.

---------------------
Post rebuttal: Thanks to the authors for the responses. After reading the response and other reviewers' comments, I'm keeping my current rating.

---

> ### Author Response · Authors · 2022-11-16
> **Thank you for your feedback!**
>
> Response to connection b/w Carlini and Feldman’s work: We thank the reviewer for making this astute observation. We raise this observation in Appendix B.1 ourselves. However, the surprising fact is that the paper by Carlini et al. does not note this connection (despite the work being published after the work of Feldman and Zhang); no subsequent work does so as well. We would happily rephrase our claims if the reviewer can point us to a reference that already claims this.
>
> In that regard, our work is the first to formally study the implication of this connection, bridging work that is published in two different communities (ML and Security). The observation being fairly intuitive is a testament to the presentation of results in our work. Because the two are deeply related, one can utilize knowledge of memorization to enhance the efficacy of MI for specific samples. This result is interesting in its own right because it is able to explain the “disparate efficacy of MI” through the lens of memorization; now we have a better understanding of the efficacy of MI for specific samples (relative to other samples that were used to train the model).
>
> Response to goodness of OOD: In the context of this work, we claim OOD-ness when there’s a mixture of two distributions. Concretely, one can visualize this as a mixture of two datasets. We would also like to point the reviewer to a more recent OOD detection method that we evaluated in Appendix D.2; the conclusions are similar to MSP. This suggests that the problem is not in the method, but the hypothesis itself.
>
> Response to game formulation: The primary aim of this paper is to show that “for those samples that are at a higher risk of being memorized, they are more susceptible to MI”. With this conclusion, we are able to shed some light on the disparate performance of MI towards different samples through the lens of memorization. One way of perceiving these results is they serve as an ablation study of understanding MI attacks.
>
> We hope you find these answers satisfactory. If so, please do consider increasing the score for the submission. If not, kindly let us know what better we can clarify.

---

### Official Review · Reviewer_HK7T · 2022-11-04

**Confidence:** 2
**Correctness:** 2
**Technical Novelty And Significance:** 2
**Empirical Novelty And Significance:** 2
**Recommendation:** 3

**Clarity, Quality, Novelty And Reproducibility:**

Contains unexplained or implicit assumptions.
Contains gaps in the logic. Due to this gap, it s difficult to examine whether it is making a valid argument adequately.

**Strength And Weaknesses:**

Strength
Theoretical considerations on the success of membership inference.

Weakness
Contains unexplained or implicit assumptions
Contains gaps in the logic. Due to this gap, it s difficult to examine whether it is making a valid argument adequately

**Summary Of The Paper:**

This study focuses on the existence of samples in which membership inference is more likely to succeed and samples in which it is less likely to succeed and discusses the causes of this difference. It was generally believed that out-of-distribution samples were more likely to succeed in membership inference, but the study argues that whether the sample is in-distribution or out-of-distribution has little to do with whether the sample is in-distribution or out-of-distribution. Instead, they argue that the larger the gap between the probability that a sample is identified as the target label when trained on a training data set that contains the target sample and the data set that does not, the more likely it is that membership inference will be successful. In addition, theoretical and experimental considerations regarding this hypothesis are provided.




**Summary Of The Review:**

The author's argument could not be fully understood because it either assumes knowledge and understanding of existing research and assumptions contained in them or because there are gaps in the logic. The paper's claims must be structured in such a way that they can be understood on their own.

Page 4
The explanation of the privacy score is too simple.

Page 5
Why do you only include samples with large memorization values in your attempt to correlate memorization and privacy scores? Isn't it impossible to evaluate the correlation unless we also include samples with small memorization values?

Figure 2, bottom
How were these image samples selected? Are they not cherry-picked?

Page 6
How were samples in D_A  drawn? Sampled from out-of-distribution?

Proof of Lemma
What does the hybrid argument mean?

Page 7
In Reason 1, do you assume \theta is optimal?
In Reason 2, do you assume the training procedure is probabilistic?

---

> ### Author Response · Authors · 2022-11-16
> **Thank you for the feedback!**
>
> Response to gaps in logic: Could the reviewer kindly clarify what gaps exist? We will provide more details to alleviate these concerns, as well as edit the paper based on this feedback.
>
> Response for privacy score definition: We utilize the same definition as Carlini et al. in https://arxiv.org/pdf/2112.03570.pdf. We will provide a more formal definition.
>
> Response to comment on page 5: Figure 5 in Appendix D2 plots the relationship b/w memorization and privacy score for all OOD points (which are plotted in red). Observe that there is no clear correlation. But when we consider those samples with a high memorization score, we can see that they also correspond to those points whose MI susceptibility is high.
>
> Response to figure 2 selection: All images are randomly chosen. The first 3 rows are images which have a low memorization score (0 or close to 0). The next 3 rows are images that have a moderate memorization score (0.5 or close to 0.5). The last 3 rows are images that have a high memorization score (1 or close to 1).
>
> Response to drawing samples to D_A: When D_A is populated solely with OOD samples, the MI attack is not particularly effective (on samples from D_A). When D_A is populated with singletons that are also OOD, the MI attack is substantially more effective. More details are presented in Table 3.
>
> Response to “hybrid argument”: The essence of the proof is to show that the difference between the advantage for a single point and a dataset with ‘m’ points is bounded. This can be obtained by calculating the difference between advantage for a single point and then 2 points, and working the way upto ‘m’ points. More details are provided in https://en.wikipedia.org/wiki/Hybrid_argument_(Cryptography).
>
> Response to comment on page 7: Through ERM, one can obtain parameters “close to” optimal, but not necessarily optimal. This is measured by the generalization gap, which captures the predictive capabilities of the model. The training procedure is randomized.
>
> We hope you find these answers satisfactory. If so, please do consider increasing the score for the submission. If not, kindly let us know what better we can clarify.

---

### Decision · Program_Chairs · 2023-01-20

**Decision:**

Reject

**Justification For Why Not Higher Score:**

See review

**Justification For Why Not Lower Score:**

N/A

**Metareview: Summary, Strengths And Weaknesses:**

This paper first aims to argue that success of membership inference attack is not necessarily related to a sample being out-of-distribution. It then argues that is related to label memorization in the sense defined in (Feldman, 2020). There are substantial issues with both of these parts. There is no perfect way of measuring how much OOD is a sample but instead a variety of proxy scores. So the fact that a certain proxy score does not capture the MI success accuracy does not actually disprove the intuition based on OOD (even if I tend to agree that this informal intuition is flawed). The more significant issue is that the connection between memorization and MI accuracy is known and rather obvious given the definitions (see review by eP6o for additional details). Is known in the privacy community and connections between memorization and MI are mentioned explicitly for example in Sec 4.3 of (Feldman, 2020)  https://arxiv.org/pdf/1906.05271.pdf